# Improvement of stomatal resistance and photosynthesis mechanism of Noah-MP-WDDM (v1.42) in simulation of $NO_2$ dry deposition velocity in forests

Ming Chang[1,*], Jiachen Cao[1,*], Qi Zhang[2,3], Weihua Chen[1], Guotong Wu[1], Liping Wu[1], Weiwen Wang[1], and Xuemei Wang[1]

[1]Guangdong-Hongkong-Macau Joint Laboratory of Collaborative Innovation for Environmental Quality, Institute for Environmental and Climate Research, Jinan University, China
[2]School of Atmospheric Sciences, Sun Yat-sen University, China
[3]Tianjin Academy of Eco-environmental Science, China
[*]These authors contributed equally to this work.

**Correspondence:** Xuemei Wang (eciwxm@jnu.edu.cn)

**Abstract.**

Rapid urbanization and economic development in China have led to a dramatic increase in nitrogen oxide ($NO_2$) emissions, causing serious atmospheric nitrogen pollution and relatively high levels of nitrogen deposition. However, despite the importance of nitrogen deposition, dry deposition processes in forested areas are still insufficiently represented in current global and regional atmospheric chemistry models, which constrains our understanding and prediction of spatial and temporal patterns of nitrogen transport in forest ecosystems in South China. The offline 1-D community Noah land surface model with multi-parameterization options (Noah-MP) is coupled with the WRF-Chem dry deposition module (WDDM) and is applied to further understand and identify the key processes that affect forest canopy dry deposition. The canopy stomatal resistance mechanism and the nitrogen-limiting scheme for photosynthesis in Noah-MP-WDDM are modified to improve the simulation of reactive nitrogen oxide dry deposition velocity. This study finds that the combined improved stomatal resistance mechanism and nitrogen-limiting scheme for photosynthesis (BN-23) agree better with the observed $NO_2$ dry deposition velocity, with mean bias reduced by 50.1%, respectively. At the same time, by comparing the different mechanisms of the two processes of canopy stomatal resistance and leaf nitrogen-limiting factors, this study also finds that the diurnal changes in dry deposition velocity simulated by each regional model present four sets of distributions. This is mainly due to the different ways that each integrated mechanism handles the opening and closing of stomata at noon and the way the nitrogen-limiting factor acts.

# 1 Introduction

Transport and deposition of nitrogen-containing compounds is one of the most critical processes in the study of biogeochemical cycles (Gruber and Galloway, 2008). Atmospheric nitrogen deposition is not only the main way that atmospheric reactive nitrogen is removed but is also an important source of nitrogen for ecosystems (Jefferies and Maron, 1997; Horii et al., 2005). Nitrogen deposition affects changes in the carbon sink in forest ecosystems by affecting plant growth and death (De Vries et al., 2009; Bernhard, 2012). An increase in nitrogen deposition will cause an increase in litter and a decrease in soil decomposition, which will increase the carbon fixation of the soil (Stevens et al., 2004; Liang et al., 2020). Meanwhile, soil acidification caused by nitrogen deposition will reduce the number of microorganisms in the soil, reduce the production of methane, cause the degradation of peatland, and jointly affect the balance of greenhouse gases and the climate (Xu et al., 2009; Cui et al., 2010; Seinfeld and Pandis, 2012; Erisman et al., 2014). At present, studies have shown that there has been a sharp rise in global and regional atmospheric nitrogen deposition, which exceeds the critical load of local ecosystems in many regions (Liu et al., 2013; Yu et al., 2019).

In order to evaluate the impact of atmospheric nitrogen dry deposition on ecosystems, it is important to accurately estimate the dry deposition fluxes of nitrogen components (Wu et al., 2011, 2012; Tian et al., 2018). Scholars calculate the dry deposition velocity of nitrogen-containing components or estimate the effect of variation on dry deposition flux based on global or regional numerical models (Phillips et al., 2006; Zhao et al., 2017; Han et al., 2017; Zhong et al., 2020). The biased results of nitrogen deposition from modelling compared to observations range from -70% to 800% (Chang et al., 2020a). Part of the estimated uncertainty comes from the input bias in the nitrogen emission inventory in the model simulation. For example, Galloway et al. (1994) predicted the nitrogen deposition pattern for 2020, and a large deflection area appeared in an area where emissions did not increase as expected. Another part of the uncertainty comes from the inaccurate simulation of nitrogen concentration. In this situation, the simulated concentration can be verified and nudged by measuring stable oxygen and nitrogen isotope ratios (Guerrieri et al., 2020). At the same time, the simplification and biases of the deposition mechanism in the models compared with satellite retrievals cannot be ignored (Liu et al., 2020). Deposition velocity is difficult to measure and is affected by many coupled physical, chemical, and biological processes occurring at the deposition interface.

Therefore, the resistance-velocity method, which is similar to Ohm's law, is used to calculate the dry deposition velocity of various atmospheric species between the atmosphere and the land surface (Szinyei, 2015). In this method, the dry deposition velocity ($V_d$) of gaseous matter is expressed as the reciprocal of the total resistance ($R_t$) of the atmospheric pollutants' deposition process to the land surface. The total resistance is determined by aerodynamic resistance($R_a$), quasi-laminar boundary layer resistance ($R_b$), and canopy resistance ($R_c$). Their relationship is generally characterized by Eq. 1.

$$V_d = \frac{1}{R_t} = \frac{1}{R_a + R_b + R_c} \tag{1}$$

Here, $R_a$ is calculated by micrometeorological parameters, which depend mainly on local atmospheric turbulence intensity, while $R_b$ is driven by the diffusion coefficient and air viscosity of gaseous matter. The calculation of these two resistances in different deposition mechanisms follows similar principles (Finnigan, 2000). At present, the treatment of dry deposition

processes affected by turbulent diffusion in numerical models includes two parts: one is the turbulent diffusion process from the bottom of the atmospheric boundary layer to the canopy, and the other is the turbulent exchange process inside the canopy (Flechard et al., 2011, 2013).

The calculation of $R_a$ is usually based on the turbulent transport part of the land surface model. Most current models are based on the near-surface-layer similarity theory, which first calculates the surface roughness and zero plane displacement,

and then calculates the turbulent transport coefficient according to the flux gradient relationship under different stratifications (Makar et al., 2017). The calculation of turbulent exchange inside the canopy is more complex and is highly related to the structure of the vegetation canopy and other local properties (Finnigan et al., 2009). Some forest fire models are based on the measured empirical wind speed profiles in the canopy, and other models use the assumption of neutral stratification to solve the turbulent flow fields of the canopy, such as SSiB, SVAT, BATS, etc. (Yongjiu and Qingcun, 1997; Yang and Friedl, 2003;

Falge et al., 2005; Moon et al., 2019).

Furthermore, the calculation of $R_c$ is more complicated and diverse than that of $R_a$, because $R_c$ is closely related to differences in the underlying surface, vegetation, soil, and other conditions (Wu et al., 2018). Due to different in the underlying surface, $R_c$ is usually further decomposed based on canopy type, canopy structure, surface properties of deposition receptors, biochemical reactions of deposition materials, mesophyll uptake, and other canopy processes (Ganzeveld et al., 2002; Wolfe

and Thornton, 2011; Simpson et al., 2012; Delaria and Cohen, 2020; Massad et al., 2020). For the surface of the vegetation canopy, models are refined to consider the resistance of the stomata, mesophyll, epidermis, soil, and other canopy surface factors (Dai et al., 2004; Massad et al., 2020). For example, a multi-layer forest canopy model is used to calculate the canopy stomatal resistance layer by layer at monitoring sites in the Clean Air Status and Trends Network (CASTNET) (Li et al., 2016). As a counterexample, the ocean, which was thought to be a relatively simple surface, has evolved from consideration of smooth

levels to sea surface fragmentation, different particle humidities, and other factors (Schulz et al., 2012). However, our current understanding of the exchange of nitrogen oxides between the atmosphere and biosphere remains incomplete; Delaria and Cohen (2020) proves the importance of $NO_2$ dry deposition and demonstrates that $NO_2$ deposition can provide a mechanistic explanation for the canopy reduction of $NO_x$, which has been ignored or unexplained by current common land surface models. For instance, the possible existence of an $NO_2$ compensation point toward the leaf surface in forests has been controversial as a

result of experimental comparison (Wang et al., 2020). At the same time, Delaria et al. (2018)'s work found that the hypothesis of a nitrogen compensation point may be a problem caused by not adopting a direct $NO_2$ measurement technique. And the interferences from alkenes or other reactions of biogenic volatile organic compounds may enhance the observed $NO_2$ compensation point and suppress the deposition velocity (Delaria et al., 2018; Place et al., 2020). This will likely lead to changes in our traditional treatment of the parameterization of nitrogen exchange in the model. The coupling of canopy photosynthesis,

nutrient stress, impact of mesophyllic processes, and other plant physiological processes is still poorly resolved in the field of dry deposition model improvement (Massad et al., 2020).

In this study, we apply different improved stomatal resistance mechanisms and nitrogen limitations on photosynthesis mechanisms to the Noah-MP model coupled with dry deposition schemes to explore changes in nutrient stress in stomatal conductance and evaluate the consequences of these changes on $NO_2$ dry deposition velocity. This paper is organized as follows:

besides this introductory Sect. 1, Sect. 2 presents a full description of the improved stomatal resistance mechanisms and different schemes of nitrogen limitation of photosynthesis. Sect. 3 includes model evaluation and discussion about the influence on $NO_2$ dry deposition velocity simulation, the respective path of canopy stomatal and photosynthesis processes, and the sensitivity of major parameters. Finally, a conclusion and a future research plan for the Noah-MP-WDDM framework are summarized in Sect. 4.

## 2 Model Description and Configuration

### 2.1 Base Model Setup

This study uses the coupled single-point (1-D) Noah-MP model and the WRF-Chem dry deposition module (WDDM) as the base model (Noah-MP-WDDM) which was developed by Zhang et al. (2017). In order to reduce the effect of meteorological simulation biases on $V_d$ simulation, the micro-meteorological observation test datasets of Zhang et al. (2017) are used to drive this dry deposition single-point (1-D) Noah-MP-WDDM model and all improvements. However, it is worth noting that when this single-point simulation is upscaled to a regional or global model, it may bring more uncertainty due to the scale conversion. In addition, all the land surface parameters used in this study are the default parameters inside the Noah-MP land surface model's look-up tables (VEGPARM.TBL, SOILPARM.TBL, and MPTABLE.TBL). This may cause systematic uncertainties in the overall modelling. The observation data were obtained at the Dinghushan Forest Ecosystem Research Station (Fluxnet Site Code: CN-Din, $23°10'24''$N,$112°32'10''$E, altitude 300m). The $NO_2$ concentration was measured using the Model T200 (Teledyne-API, USA) $NO_2$ analyzer (Zhang et al., 2017).

Physical processes related to snow, permafrost, and other factors—like Supercoiled Liquid Water in Frozen Soil (FRZ), Frozen Soil Permeability (INF), Snow Surface Albedo (ALB), Partitioning Precipitation into Rainfall and Snowfall (SNF), Lower Boundary of Soil Temperature (TBOT), and Snow/Soil Temperature Time Scheme (STC)—have only a small effect on $V_d$ because Dinghushan is located in the subtropics. So these physical parameterization schemes all use the default option (Niu, 2011). In contrast, the other six physical parameterization schemes—Dynamic Vegetation Model (DVEG); Canopy Stomatal Resistance (CRS); Soil moisture factor for stomatal resistance, $\beta$ Factor (BTR); Runoff and Groundwater (RUN); Surface Exchange Coefficient for Heat, $C_H$ (SFC); and Radiation Transfer (RAD)—have a great influence on $V_d$ simulation. Their respective options are the dynamic vegetation option (opt_dveg=2), Ball-Berry canopy stomatal resistance option (opt_crs=1), BATS soil moisture factor option (opt_btr=3), original surface and subsurface runoff option (opt_run=3), original Noah surface layer drag coefficient option (opt_sfc=2), and two-stream applied to grid-cell radiation transfer option (opt_rad=2) in the above physical parameterization schemes (Niu, 2011; Chang et al., 2020b).

### 2.2 Coupling of Stomatal Resistance Schemes

Previous studies have generally used the Jarvis stomatal conductance model, which is based on environmental factors such as photosynthetic effective radiation, temperature, humidity, and soil water to calculate canopy stomatal resistance (Jarvis,

1976). Compared with Jarvis, the Ball-Berry stomatal conductance model (Ball et al., 1987) calculates the stomatal resistance based on the through-canopy photosynthesis rate, $CO_2$ concentration, and humidity on the leaf surface as shown in (2). This type of mechanism requires a coupled photosynthesis model to calculate or observe the photosynthesis rate of the canopy, and the photosynthesis model depends on the setting of many plant physiological parameters (such as optimal photosynthesis efficiency, catalytic enzyme activity parameter Q10, etc.). It is worth noting that these parameters are often inaccurate at the regional scale, which brings some uncertainty (Dai et al., 2019; Fisher and Koven, 2020).

Although the Ball-Berry type stomatal resistance scheme behaves very similarly to the Jarvis type in modeling transpiration, the former scheme allows a direct coupling of terrestrial water and carbon fluxes and improves the simulation of vegetation–atmosphere interactions (Niyogi et al., 2009; Yang et al., 2011). The Noah-MP model sets the stomatal conductance slope of the Ball-Berry mechanism as a constant, which is not suitable and will cause a large simulation bias. Therefore, we integrate observational experimental results, statistical fitting or plant physiological model equations in photosynthesis, stomatal conductance, and other aspects of plant physiology in this study, by writing the equation as subroutines and adding to the calling tree in the coupled single-point Noah-MP-WDDM model.

The calculation equation is Ball et al. (1987) as follows:

$$\frac{1}{R_s} = m \times \frac{A}{C_{air}} \times \frac{e_{air}}{e_{sat}(T_v)} \times P_{air} + g_{min} \tag{2}$$

Where $m$ is the slope of the stomatal conductance, $A$ is the photosynthetic rate, $C_{air}$ is the $CO_2$ concentration on the leaf surface, $e_{air}$ is the vapor pressure on the leaf surface, $sat(T_v)$ is the saturated vapor pressure of the leaves at the canopy temperature, $P_{air}$ is the surface pressure, and $g_{min}$ is the minimum stomatal conductance.

In addition, the non-stomatal resistance ($R_{ns}$) calculated in Noah-MP is according to (Zhang et al., 2003):

$$\frac{1}{R_{ns}} = \frac{1}{R_{ac} + R_g} + \frac{1}{R_{cut}} \tag{3}$$

where $R_{ac}$ is the in-canopy aerodynamic resistance, which is common to all gases, and $R_g$ and $R_{cut}$ are the resistances for the uptake by the ground/soil and canopy cuticle. Similar to the work of Wesely (1989), $R_g$ and $R_{cut}$ are parameterized for $O_3$ from look-up tables.

The equations integrated into the single-point mechanism model are shown in Table 1, where

- MBM-1 (Modified Ball-Berry Mechanism, MBM) is the stomatal conductance equation of the default Ball-Berry equation, and the main parameter used is the slope of the Ball-Berry conductance relationship and the minimum stomatal conductance ($g_{min}$);

- Leuning (1990) introduced a $CO_2$ compensation point $\Gamma$ to improve the Ball-Berry equation so that it can simulate the net photosynthetic rate and stomatal conductance when the $CO_2$ concentration on the blade surface is equal to the compensation point (Table 1, MBM-2). And the method of Lohammar et al. (1980) was adopted to replace $RH$ with

the water vapor saturation function $f(D)$ while its equation has been applied to a variety of plant physiological models (Leuning, 1995) (Table 1, MBM-4);

– Aphalo and Jarvis (1993) separated the effects of temperature and water vapor difference $D$, which more directly reflects the effect of temperature on stomatal conductance than the original Ball-Berry equation (Table 1, MBM-3);

– Yu et al. (2004) measured stomatal conductance of wheat under normal atmospheric and artificially increased $CO_2$ concentration, as well as the response curve of photosynthesis to light and $CO_2$ concentration. Based on this, researchers constructed an equation reflecting the physiological response of plants, which could reflect the relationship between stomatal conductance and photosynthesis rate (Table 1, MBM-5);

– Ye and Yu (2008) derived a model of leaf stomatal mechanism based on experimental observation of light response and
stomatal conductance data, which can better simulate the relationship between stomatal conductance and photosynthetic rate (Table 1, MBM-6);

– Medlyn et al. (2011) introduced the optical contract rate coefficient $g_1$ (Table 1, MBM-7), which led to better simulation results in models such as CABLE (De Kauwe et al., 2014).

## 2.3   Improvements in Nitrogen-Limiting Schemes for Photosynthesis

The combination of the DVEG mechanism and the Ball-Berry model can comprehensively consider the interaction between photosynthesis rate and canopy stomatal resistance. This is physiologically significant in that it balances the supply and demand of $CO_2$ in the chemical reaction of photosynthesis, so as to maintain a reasonable concentration of $CO_2$ in the mesophyll tissue. However, at the vegetation canopy scale, the photosynthetic rate is also related to the nitrogen content of leaves. Currently, the commonly used biogeochemical models usually express the effect of nitrogen on the photosynthetic process based on the
relevant theory of nitrogen limitation, but they often simplify it (Li et al., 2013).

In this study, the original DVEG mechanism of Noah-MP set the nitrogen limitation factor of leaves $f(N)$ as a function of leaf nitrogen concentration ($C_N^{leaf}$) and the maximum nitrogen concentration parameters (FOLNMX) of this vegetation type ($f(N) = C_N^{leaf} \cdot FOLNMX^{-1}$). However, $C_N^{leaf}$ and FOLNMX were set as two constants which is obviously an over-simplification for land surface simulation in a large area (Bonan, 1995). For different types of plants, the nitrogen content in
leaves should make a significant difference in photosynthetic nitrogen utilization efficiency (Zheng and Shangguan, 2007). For regional nitrogen deposition simulation, it is obviously inappropriate to simplify the nitrogen-limiting process in leaves, so a more accurate description of the effect of nitrogen on plant photosynthesis and a more accurate estimation of the effect of nitrogen deposition on the whole forest ecosystem are needed.

According to whether the nitrogen content of plant tissues is directly taken as the variable in the equation, the current
expressions of how nitrogen affects photosynthesis (as shown in Table 2) can be divided into implicit and explicit expressions:

– Implicit

For the photosynthetic rate model calculated by the Farquhar Model (Farquhar et al., 1980), the photosynthetic rate is determined by the minimum value of carboxylation efficiency ($W_c$), carboxylation efficiency ($W_j$) and organophosphorus carboxylation efficiency ($W_e$), which is limited by the concentration of chlorophyll photoenzyme (Rubisco), in which $W_c$ and $W_e$ are proportional to the maximum carboxylation rate ($V_{cmax}$).

Therefore, the effect of nitrogen on photosynthesis, $V_{cmax}$, is reflected mainly in the limitation of $f(N)$. As mentioned above, $f(N)$ was set by two constants in the DVEG dynamic vegetation process mechanism. In addition, models such as AVIM (Ji, 1995), CLM4.0 (Oleson et al., 2010) and Noah–LSM (Bonan, 1995) also directly take $f(N)$ as a parameter, ranging from 0.5 to 1(Table 2, MNM-1, (Modified Nitrogen Mechanism, MNM)).

However, BEPS (Liu et al., 1999), DLEM (Tian et al., 2010), IBIS (Liu et al., 2005), InTEC (Chen et al., 2000) and other models use the ratio of the optimal carbon-nitrogen ratio ($B_V max$) to the simulated actual carbon-nitrogen ratio ($B_L$) to represent $f(N)$ (Table 2, MNM-2).

Models that calculate photosynthesis processes based on empirical functions (such as CASA (Friedlingstein et al., 1999), Lin(Lin et al., 2000), PnET(Aber and Federer, 1992), TEM(McGuire et al., 1997), TRIPLEX (Peng et al., 2002), and 3-PG (Landsberg and Waring, 1997), etc.) mostly use the form of vegetation productivity, which is proportional to light interception, to calculate the primary productivity (NPP) or total primary productivity (GPP) of vegetation (Monteith, 1972; Monteith and Moss, 1977). Such models generally use implicit methods to limit GPP or NPP, so as to implicitly limit the calculation of photosynthesis rate, thus affecting canopy stomatal conductance (Table 2, MNM-3, MNM-4).

– Explicit

The explicit method is in direct accordance with plant physiology experiments to establish the relation between the $V_{cm}$ and $c_N$ functions, and leaf nitrogen content can be measured more directly in the plant physiology sense in relation to photosynthesis. Different researchers get different function relations, so there is no unified explicit expression of the equation:

For example, in Biome-BGC (Thornton et al., 2002), $V_{cm}$ is calculated by the carbon and nitrogen ratio ($C : N_{leaf}$), the ratio of the Rubisco enzyme middle nitrogen content to total leaf nitrogen content ($f_{nr}$), specific leaf area index ($SLA$), etc. (Table 2, MNM-5);

Doly and CEVSA established a functional relationship between light saturation rate ($A_b$) and leaf nitrogen absorption rate ($n$) (Woodward et al., 1995; Cao and Woodward, 1998)(Table 2, MNM-6).

## 2.4 Experiment Setup for Mechanism Comparison

After integrating all the improved equations of the canopy stomatal resistance mechanism and the nitrogen-limiting schemes for photosynthesis into the single-point model in the form of subroutines, an orthogonal experimental scheme was adopted to simulate them, and all the experimental schemes were driven by the same meteorological forcing data. The code names of each simulation experiment are shown in Fig. 2 in which the original Noah-MP-WDDM model from Zhang et al. (2017) is named

BN-11. Since all the mechanisms can be combined into 42 combinations, the current version number is set at v1.42 in this study.

## 3 Results

### 3.1 Model Validation

To evaluate the applicability of the single-point Noah-MP-WDDM dry deposition model and all its improvements, we compared the base model results (BN-11) to the observations of latent heat (LH) and sensible heat (SH) fluxes. Detailed statistics of the comparison are shown in Table 3. It can be seen that the simulation of average SH is overestimated by about $20\ W \cdot m^2$ while the average LH is underestimated by about $0.1\ W \cdot m^2$ compared with observations. The models perform reasonably well for most simulations, with uncertainties within a factor of $0.5 \sim 2$ (Fig. 1).

### 3.2 Performance of Vd Simulation with Different Mechanisms

The model simulation shows obvious underestimation of $V_d$. The simulated average $V_d$ is about a quarter of the observed results. And the correlation coefficient is very low and basically cannot reflect the trend characteristics (Fig. 2). On the one hand, the Noah-MP-WDDM model itself has a poor ability to simulate the change trend of deposition. On the other hand, it is also affected by too much precipitation in subtropical regions, poor quality control of dry deposition observation data and many missing values (Zhang et al., 2017). The observation instrument was limited by the conditions surrounding the flux tower, and the assayed gas had accumulated (especially at night) in the reaction chamber, resulting in a partial (nocturnal) high observed value(Zhang et al., 2017).

It can be seen from Fig. 2 that the simulation effect of each model mechanism is relatively poor, especially for all the combinations corresponding to the MBM-5, MBM-6, MBM-7, and MNM-5 series; the simulated $V_d$ in these series is basically concentrated around 0.05 cm/s. This indicates that the stability of the parameterization of these series of mechanisms is relatively high, and the disturbance caused by different schemes in other processes is suppressed.

There is a magnitude difference between the results of the simulation and the observed $V_d$, which may be because these mechanisms are not supported with some coniferous species because conifers have little direct stomatal response to elevated $CO_2$ (Medlyn et al., 2011; Katul et al., 2012). Especially for the current version of the single-point Noah-MP-WDDM model, the concentration of $CO_2$ is input to the model as a parameter, which may restrict the simulation performance of the model itself. When Noah-MP-WDDM is coupled to climate or atmospheric models, it may create new sources of uncertainty. The overall underestimation under MNM-5 may be because the default parameters of the leaf carbon-nitrogen ratio ($C : N_{leaf}$) in the Biome-BGC model and single-point Noah-MP-WDDM model do not match the situation of subtropical forests.

However, we can still see the variation of the simulation bias caused by different mechanisms from the statistical results (Fig. 3). Most of the simulated combinations underestimate the average dry deposition velocity, but only three mechanism combinations—BN-16, BN-26, and BN-36—overestimate it. It can be seen that the average simulation bias of BN-23 is the

lowest among all mechanism combinations. Compared with the default BN-11 mechanism, its average bias is reduced from -0.0371 cm/s to -0.0185 cm/s, which reduces the relative deviation about 50.1%. At the same time, BN-13 and BN-33 achieve similar results, with a simulation bias of the dry deposition velocity of -0.0187 cm/s. Then for BN-46, the bias of dry deposition velocity is -0.0256 cm/s.

### 3.3  Implications for Diurnal Simulation of $NO_2$ Dry Deposition Velocity

Although the ability of the models to simulate trends is statistically weak, and the absolute difference in the average dry deposition velocity obtained from simultaneous results is small, we can still see that the model captures certain dry deposition characteristics from the daily cycle changes. Fig. 4 uses a daily variation curve to show the simulation results of the effects of each mechanism combination on the dry deposition velocity. It can be seen that for the daily variation of $NO_2$ dry deposition velocity, different mechanisms still show considerable pattern differences.

The red line in Fig. 4 is the daily change in the observed value. Note the large fluctuation of its standard deviation, indicating a large fluctuation during the $V_d$ observation period. This is because of the turbulent exchange caused by the fragmentation of the boundary layer inside and outside the mountain forest canopy and the effect that the change in atmospheric stability has on the turbulence (Zhang et al., 2017). The black line in Fig. 4 corresponds to experiment BN-11 and the green line to BN-23, respectively. It can be seen that compared to the original BN-11, the simulated Vd values of BN-23 are increased mainly during

the day. At the same time, it can be seen that the standard deviation range of the observed values can basically cover the range of the simulated results. This could partly reflect the stability of the model, which may mean that these improved mechanisms can show similar performance when transplanted to other similar types of forests.

In addition, it is worth noting that some deposition observation studies believe that the $V_d$ value at midday is the most noteworthy (Kavassalis and Murphy, 2017; Ke et al., 2020). Therefore, we also pay attention to the mechanism with the

minimum bias at midday, which is BN-46. It can be seen that the simulated value of BN-46 is basically consistent with the $V_d$ observation, with a bias of about 0.001 cm/s at midday.

Overall, the simulation of the daily variation of $V_d$ presents four groups: the greatly underestimated group (represented by BN-55), the greatly overestimated group (represented by BN-26), the morning-higher and afternoon-lower pattern group (represented by BN-23), and the accurate-at-noon group (represented by BN-46). The original Noah-MP-WDDM (BN-11)

belongs to the same group as BN-23 because their theoretical assumptions are consistent. The appearance of this grouping is quite interesting, because it illustrates that there are relative differences in theoretical assumptions about stomatal resistance and nitrogen limits for photosynthesis. Therefore, in the next section we will discuss the effects of these improved scheme groups from the perspective of canopy deposition resistances with the four representative combinations and the Noah-MP-WDDM default combination (BN-11).

### 3.4 Comparison of Modeled Resistance Components

#### 3.4.1 Aerodynamic and Quasi-laminar Boundary Layer Resistance

It can be seen that different mechanism improvements have relatively little impact on aerodynamic resistance ($R_a$) and quasi-laminar boundary layer resistance ($R_b$) since the improved mechanisms are concentrated in the canopy process. The four combinations of BN-11, BN-23, BN-26, and BN-46 are basically the same except BN-55, shown in Fig. 5 and 6. The differences in $R_a$ between BN-55 and the other four combinations are present mainly during the night, about 30 s/m, while the differences in $R_b$ range about 5~10 s/m during both the day and night.

But the source of this difference for $R_a$ and $R_b$ is slightly different. The disturbance to $R_a$ is indirectly caused by the calculation of Mourning-Obukhov length ($L$) and friction velocity ($u^*$) in the calculation of turbulence by the sensible and latent heat flux exchange controlled by the canopy stomatal mechanism. The disturbance of $R_b$ is only indirectly affected by the calculation of $u^*$.

In addition, it is worth noting that the diurnal variation of $R_b$ is more consistent with the observed $V_d$, which echoes the hypothesis of turbulence exchange caused by the breakage of the inner and outer boundary layers of the mountain forest canopy and changes in atmospheric stability proposed by Zhang et al. (2017). The model needs to express these situations by accurately expressing the forest structure.

#### 3.4.2 Canopy Resistance

The difference in the simulation of the canopy resistance ($R_c$) of each improvement scheme is the main source of the difference in the simulations of the dry deposition process, as shown in Fig. 7. It can be seen that for the consistently underestimated group represented by BN-55, the underestimation of deposition velocity comes from a large overestimation of $R_c$, indicating that the assumptions of these mechanisms are not suitable for subtropical forests as a whole. It is possible to draw inappropriate conclusions on such underlying surfaces from the source model results.

For BN-11 and BN-23, it can be seen that their theoretical hypotheses are the same, which can effectively reflect the physiological process of the increase in stomatal resistance caused by the closure of stomata at noon. The degree of stomatal reopening in the afternoon is slightly weaker, which makes the diurnal dry deposition velocity curve high in the morning and low in the afternoon. This is correct in a general sense, but there is a certain mismatch between the simulation and the observed results (low in the morning and high in the afternoon). We estimate that because the flux tower of the sample site of Dinghushan is located on a westward slope, the physiological activity of the vegetation canopy is weaker in the morning and stronger in the afternoon. This indicates that for model improvement, the parameterization of the difference between sunlit and shaded leaves should be strengthened; otherwise it will be difficult to express this phenomenon.

It can be seen that the theoretical hypotheses for BN-26 and BN-46 are also the same, but neither can reflect the closure of stomata at noon. The difference is reflected mainly in the intensity of the decrease in $R_c$ during the day, and the amplitude of the disturbance to the deposition velocity is greatly enhanced when the deposition resistance is lower than 1000 s/m. It can be seen from Fig. 4 and Fig. 7 that the difference in the deposition velocity curves obtained by the simulation of BN-26 and

BN-46 is not yet apparent at 07:00 in the morning. At 12:00, the difference in canopy resistance, which is only about 200 s/m lower, causes the BN-26 group to greatly overestimate deposition velocity. Under this series of mechanisms, the misestimation or disturbance of key parameters is likely to change the expected results.

## 4 Discussion

We improved upon the early Noah-MP-WDDM version and our results emphasize the importance of the canopy stomatal carbon dioxide compensation mechanism and GPP-controlled leaf nitrogen-limiting factor for the simulation of nitrogen deposition are overstated. All the classic model mechanisms do a fairly poor job at capturing the deposition velocity of reactive nitrogen at the chosen site, which indicates that there are substantial gaps in our current understanding and parameterization of in-canopy processes.

From a model point of view, some articles have begun to consider the internal processes of the canopy. For example, some models divide the process in the canopy into multiple layers, trying to distribute the radiation energy and the profile in the canopy more accurately, but it is only divided into non-stomatal and stomatal pathways for the material exchange process in the canopy, while the parameterization of non-stomata is often set to various empirical constants (Bonan et al., 2021). There are some experiments that assume that the process of plant surface may not be as simple as we thought in the past, so different parameters are set for the wet surface and the dry surface (Jia et al., 2016). However, the processes were just parameterzation using constants lacking physical meaning in different surface conditions. Recent isotope observational evidence also shows that forest canopies can retain nitrogen from atmospheric deposition and the canopy N processing could alter the N supply and photosynthesis of the leaf in the short term (Wang et al., 2021a). For the simulation of non-stomatal processes on the surface of plants, more in-depth observations or open-top air chamber experiments are needed to support (Fisher and Koven, 2020).

At the same time, for the stomatal pathway, whether it is the Javis scheme or the Ball-Berry scheme, in fact, it is derived from the empirical fitting in the field of atmospheric environmental research (Schwede et al., 2011). Although it has a classical photosynthesis mechanism for C3 and C4 plants separately, the parametric description of models basically does not consider the physiological response process and environmental adaptation process of the plant from the level of the plant's own gene control mechanism (Liang et al., 2020; Durand et al., 2021). In addition, the improvement of canopy structure measurement based on technologies such as lidar also requires corresponding parameterized simulation work to improve the characterization of leaf morphological parameters in the canopy (Braghiere et al., 2021; Wang et al., 2021b). It is unrealistic to consider such in-depth consideration in the land surface model at this stage, but we believe that the consideration of biological physiological processes needs to be continuously refined while the understanding of the land-atmosphere exchange process in the ecosystem grows. For example, the biogeochemical processes such as under-canopy diffusion process of reactive nitrogen oxides, the emission of volatile organic compounds from low-canopy vegetation under nitrogen stress, the photochemical reaction through the canopy gap, the emission of soil nitrogen components, the coupling process of carbon and nitrogen ratio, etc (Weathers et al., 2001; Finnigan et al., 2009; Flechard et al., 2013; Dentener et al., 2014; Erisman et al., 2014; Makar et al., 2017; Moon

et al., 2019; Guerrieri et al., 2020; Ke et al., 2020; Wang et al., 2021a). These all have effects on the exchange of nitrogen oxides at the interface inside the canopy, and are worthy of parameterization research.

We also compared the observed dry deposition velocities results carried out on different underlying surfaces, and the simulated results using the different deposition resistance mechanisms coupled in this work, as shown in Table 4. It can be seen that the dry deposition velocities of $NO_2$ range obtained by most of the model results is basically lower than the observed

value obtained by the eddy correlation method, which is relatively consistent with the performance of most of the mechanism simulation results in this study. It can also be inferred from this that most of the regional models that adopt the default Wesely deposition mechanism, such as WRF-Chem, CMAQ and other widely used models, may underestimate the dry nitrogen deposition flux (Chang et al., 2020a). The potential impact of this underestimation deserves in-depth discussion by the entire nitrogen deposition research community.

The sources of simulation uncertainty in this study may mainly come from the following aspects: First is the lack of observational data. Although the observational data used in this research has supported the publication of related articles in the previous period, the overall data quality is not good (Zhang et al., 2017; Chang et al., 2020b). The observational conditions in subtropical forests make it difficult to set up a long-term observation in nature reserves (Tian et al., 2018). Meanwhile, the observation needs to eliminate many interference factors in the measurement, so that there are fewer data that meet the quality

requirements in the end (Xu et al., 2015). Second, this study used the Noah-MP's default lookup tables in terms of model input parameters. The parameters are not localized in this study. Quite a few parameters in them are empirical values, the average value of large-scale remote sensing, or the average value of similar underlying surfaces (Niu et al., 2011; Dai et al., 2019; Massad et al., 2020). In this regard, we believe that it is necessary to further carry out the measurement and accurate characterization of model parameters, especially vegetation canopy parameters and soil parameters, in order to further effectively reduce

the uncertainty of the simulation, and to more clearly analyze the different effects of deposition resistance mechanisms.

## 5 Conclusions

In using Noah-MP-WDDM to study dry deposition processes, we implemented new features and applied several corrections to the code. Compared to Noah-MP-WDDM v1, the improvement of the canopy stomatal resistance mechanism and the nitrogen-limiting schemes in Noah-MP-WDDM v1.42 gives new options for simulating nitrogen dry deposition velocity. Our discussion

shows that the major source of the difference in the simulations of the dry deposition process is the difference in the simulation of the Rc of each improvement scheme. The canopy stomatal and leaf nitrogen-limiting mechanisms from various classic models cannot well express the diurnal changes in leaf canopy resistance, especially the underestimation in the daytime, and present four sets of distributions, by the combination of the Yu et al. (2004) and Thornton et al. (2002) mechanisms (BN-55) and the effect of the Cao and Woodward (1998) mechanism on stomatal closure (MNM-6) at noon. This may be a source of

bias in the simulation of nitrogen deposition flux by these mechanisms' source models.

Our results emphasize the importance of the canopy stomatal carbon dioxide compensation mechanism and the GPP-controlled leaf nitrogen-limiting factor for the simulation of nitrogen deposition. Considering the combination of these two

mechanisms (BN-23 schemes in Noah-MP-WDDM v1.42 instead of Noah-MP-WDDM v1), it reduced the average simulation bias by about 50.1%.

Our future work will focus on applying the combination of these mechanisms to the regional and global Noah-MP-WDDM model to simulate dry deposition for other surface types and other components. We hope to gain a deeper understanding of the simulation performance of canopy stomatal and leaf nitrogen-limiting mechanisms for dry deposition to learn more about the response and feedback of ecosystems and nitrogen deposition.

*Code and data availability.* The current version of model is available from the project website: https://zenodo.org/record/4756246 under the
Creative Commons Attribution 4.0 International license. The exact version of the model used to produce the results used in this paper is archived on Zenodo (10.5281/zenodo.4756246), and all the simulation results are presented on Zenodo (10.5281/zenodo.4756316).

*Author contributions.* MC designed the research, conducted the model development, and drafted the paper. JC performed the model simulation and drafted the paper. QZ and WC did the Noah-MP-WDDM code review. GW and LW did the model output analysis. WW did the polish work. MC was supervised directly by XW during the model development work. All authors contributed to the interpretation of the
results.

*Competing interests.* The authors declare that they have no conflict of interest.

*Acknowledgements.* This research was supported by the National Key Research and Development Plan (2017YFC0210103), National Natural Science Foundation (41705123, 41905086), Special Fund Project for Science and Technology Innovation Strategy of Guangdong Province (Grant No.2019B121205004), and Guangdong Innovative and Entrepreneurial Research Team Program (2016ZT06N263). Calculation for
this work were supported by The High Performance Public Computing Service Platform of Jinan University. We thank Ms. Laurel Anderton for her linguistic assistance during the preparation of this manuscript.

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

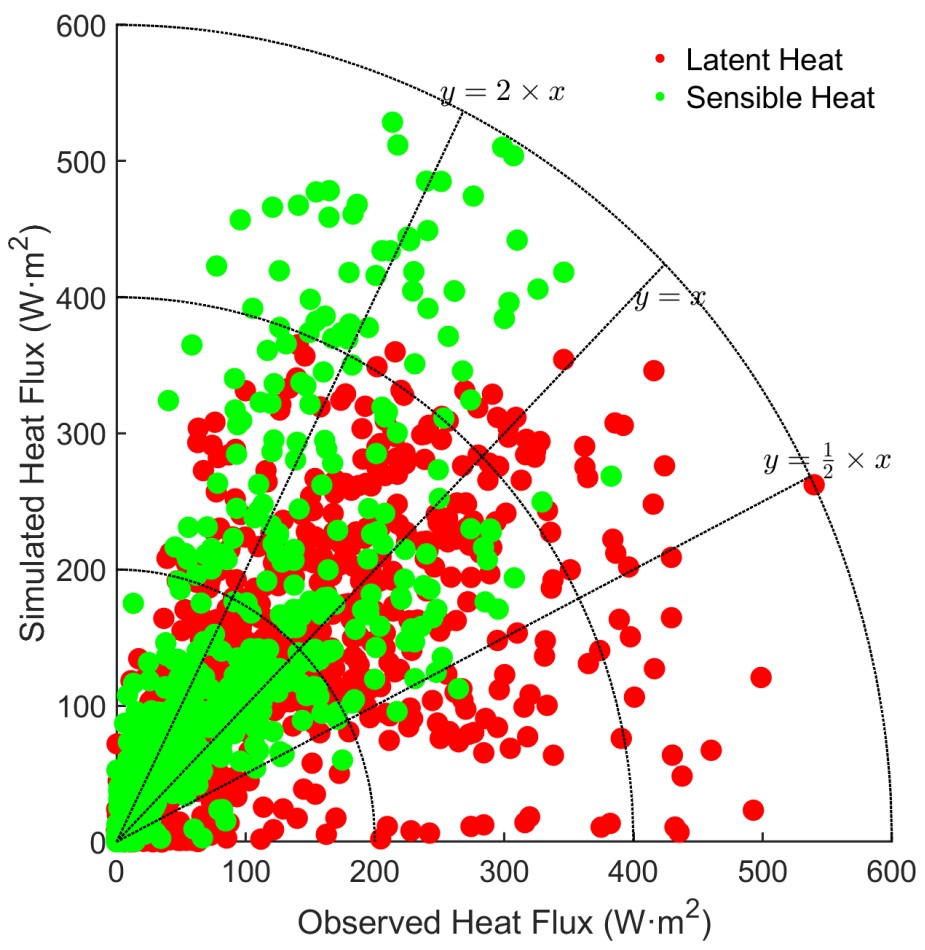

**Figure 1.** Comparison of observed and simulated fluxes of latent heat and sensible heat

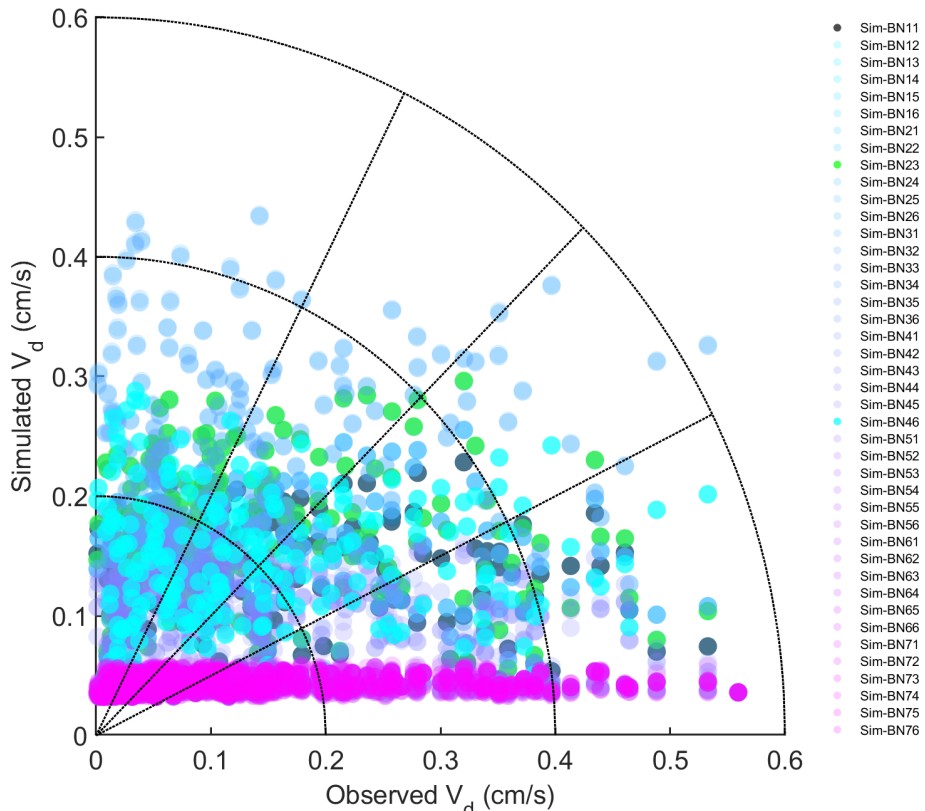

**Figure 2.** Comparison of observed and simulated $V_d$ of $NO_2$

**Table 1.** The coupled stomatal conductance and resistance equation lists

| Experiment code | Mechanism equation* | Reference |
|---|---|---|
| MBM-1 | $g_s = \dfrac{1}{R_s} = m \times \dfrac{A}{C_{air}} \times \dfrac{e_{air}}{e_{sat}(T_v)} \times P_{air} + g_{min}$ | Ball et al. (1987) |
| MBM-2 | $g_s = g_0 + a \cdot A_n \dfrac{RH}{C_s - \Gamma^*}$ | Leuning (1990) |
| MBM-3 | $g_s = \dfrac{A_n \times RH}{C_s} [k_0 + k_1 \times D + k_2 \times T_1 + k_3 \times T_1 \times D]$ | Aphalo and Jarvis (1993) |
| MBM-4 | $g_s = g_0 + a \cdot \dfrac{An}{(1 + \dfrac{D}{D_0}) \cdot (C_s - \Gamma^*)}$ | Leuning (1995) |
| MBM-5 | $g_s = a \cdot \dfrac{V_{cmax}\alpha Q\eta}{V_{cmax}\alpha Q + V_{cmax}\eta C_a + \alpha Q\eta C_a} \cdot \dfrac{1}{1 + D/D_0} \cdot \dfrac{\psi - \psi_0}{\psi_m - \psi_0}$ | Yu et al. (2004) |
| MBM-6 | $g_s = g_0 + \dfrac{1}{4\eta} \cdot \dfrac{A_{net}}{C_a - C_i} \cdot f(RH, T_L)$ | Ye and Yu (2008) |
| MBM-7 | $g_s = g_0 + 1.6 \cdot (1 + \dfrac{g_1\beta}{\sqrt{D}})\dfrac{A}{C_s}$ | Medlyn et al. (2011) |

*, For the symbols in the mechanism equations, please refer to the source literature

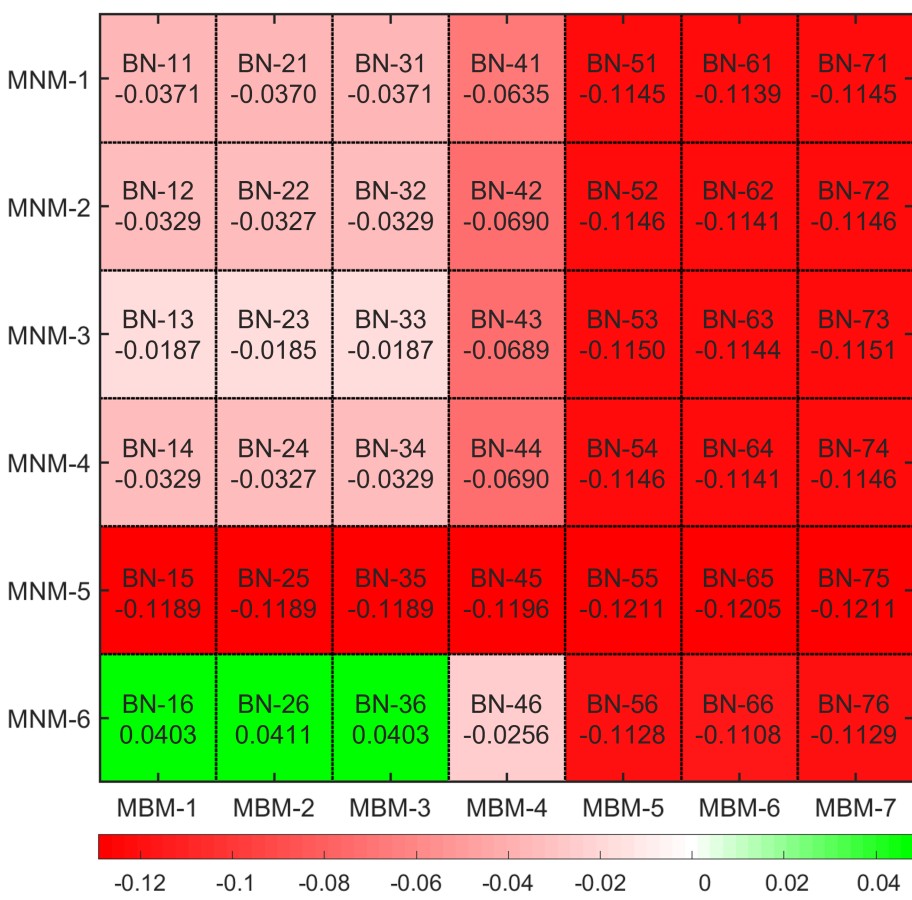

**Figure 3.** Mean bias of observed and simulated $V_d$

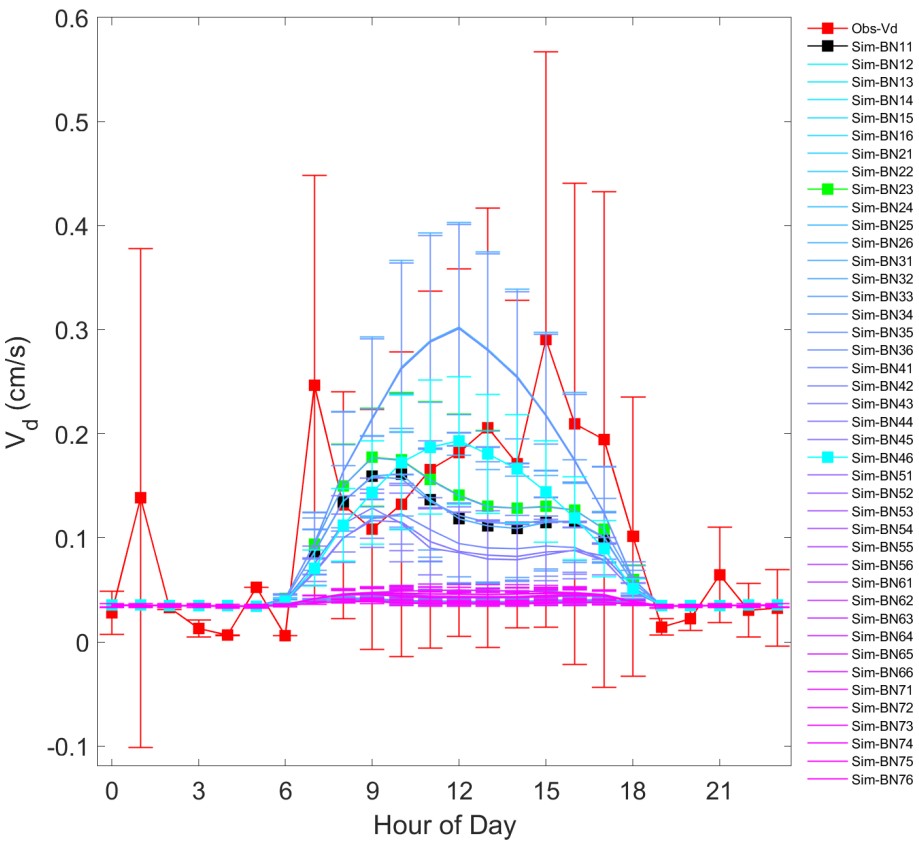

**Figure 4.** Diurnal variation of observed and simulated $V_d$

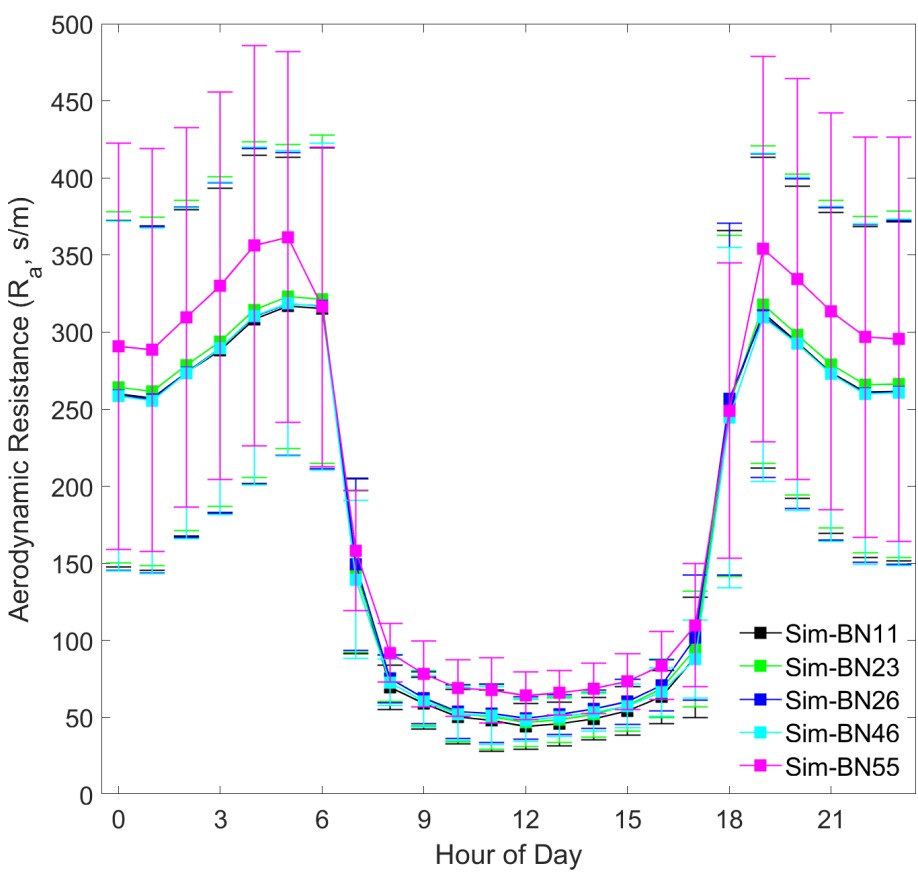

**Figure 5.** Diurnal variation of simulated $R_a$

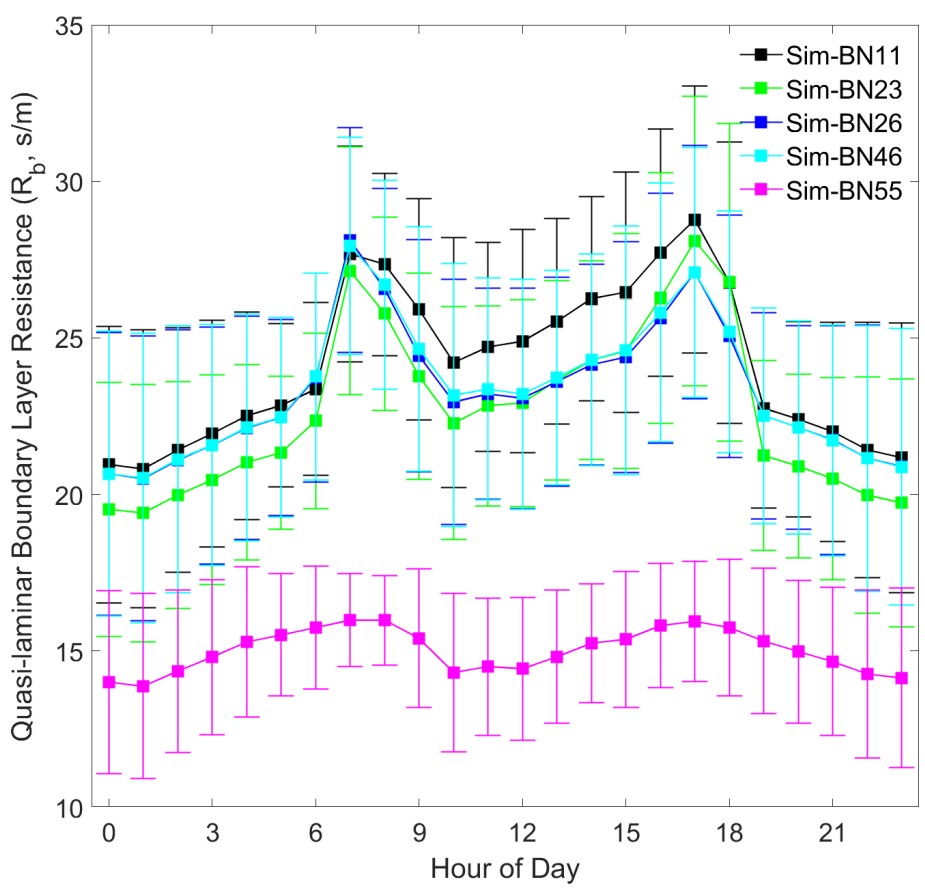

**Figure 6.** Diurnal variation of simulated $R_b$

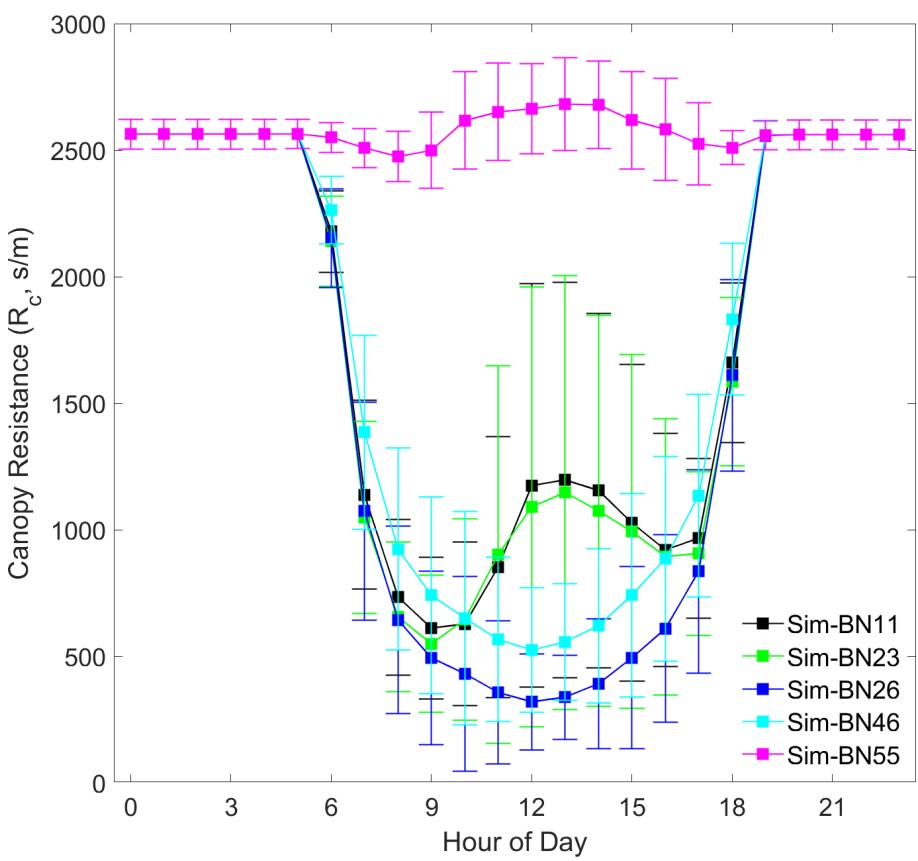

**Figure 7.** Diurnal variation of simulated $R_c$

**Table 2.** Nitrogen limits schemes for the photosynthesis mechanism

| Experiment code | Classify | Mechanism equation* | Source models |
|---|---|---|---|
| MNM-1 | Implicit | $v_{cm} = f(N) \cdot V_{cmax}$ <br> $f(N) \in [0,1]$ | Noah-MP, Noah-LSM <br> CLM4.0, AVMIN |
| MNM-2 | Implicit | $v_{cm} = f(N) \cdot V_{cmax}$ <br> $f(N) = B_L \cdot B_{Vmax}$ | IBIS, InTEC <br> BEPS, DLEM |
| MNM-3 | Implicit | $GPP = f_{GPP,N} \cdot f_{GPP,others} \cdot GPP_{max}$ <br> $f(GPP,N) = \frac{N_{av}}{k+N_{av}}$ | Lin et al. (2000) |
| MNM-4 | Implicit | $GPP = f_{GPP,N} \cdot f_{GPP,others} \cdot GPP_{max}$ <br> $f(GPP,N) = min(1.0, \frac{N_{av}}{\frac{NPP_{max}}{B_{max}}})$ | TRIRLEX, 3-PG |
| MNM-5 | Explicit | $v_{cm} = \frac{act \cdot f_{lnr}}{f_{inr} \cdot SLA \cdot C:N_{leaf}}$ | Biome-BGC |
| MNM-6 | Explicit | $v_{cm} = \frac{(A_b + R_d[P_c + K_c(1+\frac{P_0}{K_0})])}{P_c - 0.5P_0/\tau}$ <br> $A_b = \frac{190 \cdot n}{360+n}$ | CEVSA, Doly |

*, For the symbols in the mechanism equations, please refer to the source literature

**Table 3.** Statistical results of original simulated values and observed values

| Evaluation criteria | Latent heat flux | Sensible heat flux |
|---|---|---|
| Mean Observation | 66.42 | 29.81 |
| Mean Simulation | 65.20 | 48.55 |
| Mean Bias | -0.14 | 20.18 |
| Mean Absolute Percentage Error (MAPE) | 2.36 | 3.12 |
| Root Mean Square Error (RMSE) | 82.29 | 60.64 |
| Correlation Coefficient (R) | 0.68 | 0.84 |

**Table 4.** Comparison of NO$_2$ dry deposition velocities with some other studies

| Station | Land surface type | Method | Vd (cm/s) | Reference |
|---|---|---|---|---|
| Leende, NL | Grassland | Eddy Correlation Method | 0.1~0.35 | Coe and Gallagher (1992) |
| Elspeetsche, NL | Grassland | Gradient Method | 0.1~0.4 | Erisman et al. (1994) |
| Harvard Forest, USA | Coniferous Forests | Eddy Correlation Method | 0.60~0.86 | Wu et al. (2011, 2012) |
| California, USA | Laboratory Measurements | Laser-induced Fluorescence Detection Method | 0.15~0.51 | Delaria and Cohen (2020) |
| Hohenpeißenberg, DE | Coniferous Forests | Eddy Correlation Method | 0.01~0.45 | Stella et al. (2013) |
| Birmingham, UK | Cropland | Inferential Method (Wesely scheme) | 0.16 | Marner and Harrison (2004) |
| EMEP monitors | Forest | Inferential Method (Wesely scheme) | 0.15 | Flechard et al. (2011) |
| .. | Grassland | Inferential Method | 0.12 | .. |
| .. | Cropland | Inferential Method | 0.10 | .. |
| CAPMON monitors | Coniferous Forests | Inferential Method (Zhang scheme) | 0.16~0.28 | Zhang et al. (2009) |
| ... | Broadleaf Forests | Inferential Method | 0.13 | ... |
| ... | Grassland | Inferential Method | 0.11~0.22 | ... |
| ... | Cropland | Inferential Method | 0.07 | ... |
| Northern China | Forests | Inferential Method (Wesely scheme) | 0.02~0.09 | Pan et al. (2012) |
| Central Africa | Forests | Inferential Method (Wesely scheme) | 0.31~0.33 | Adon et al. (2013) |