# Peer review of "Improvement of stomatal resistance and photosynthesis mechanism of Noah-MP-WDDM (v1.42) in simulation of $NO_2$ dry deposition velocity in forests"

_Geoscientific Model Development, 2021_

## Author Response (AR1)

**Reply to 'Comment on gmd-2021-157'**

August 24, 2021

Dear Editor, Thank you for your careful review of our manuscript. We appreciate all the referee for their insightful comments and constructive suggestions. These comments and suggestion are all very helpful for improving our manuscript. We have considered all the points as discussed below and have revised the manuscript accordingly. The changes have been highlighted in blue and red fonts in the revised version by using the LaTeXdiff tool. We sincerely hope that the manuscript in its revised form will satisfy the queries of the referee and will be accepted for publication.

The comments from referees are colored black.
The authors' replies are colored blue.

**RC1: 'Comment on gmd-2021-157', Anonymous Referee #1, 05 Jul 2021**

Improvement of stomatal resistance and photosynthesis mechanism of Noah-MP-WDDM (v1.42) in simulation of $NO_2$ dry deposition velocity in forests presents results of different model mechanisms for representing stomatal deposition of $NO_2$. The authors conclude that they substantially improved upon the earlier Noah-MP-WDDM version and also assert that canopy stomata and leaf nitrogen-limiting mechanisms from various classic models cannot well express the diurnal changes in stomatal deposition.

This work is interesting and I recommend publication only following a reworking of the scope and conclusions of the manuscript. In my opinion

the conclusions that the authors have substantially improved upon the earlier Noah-MP-WDDM version and that the results emphasize importance of the canopy stomatal carbon dioxide compensation mechanism and the GPP-controlled leaf nitrogen-limiting factor for the simulation of nitrogen deposition are overstated. The more interesting finding is that all classic model mechanisms do a fairly bad job at capturing the stomatal deposition of reactive nitrogen at the chosen site in China, indicating there are substantial gaps in our current understanding of in-canopy processes. This paper would be substantially improved with an expanded discussion of what those knowledge gaps are, and how this current study is able to identify areas where more research is needed. Although I think certain findings of Chang et al., are significant, they should also be better placed in the context of recent publications. I would like to see more discussion of the results of this paper in comparison with other models, observations, and laboratory findings. The lack of discussion as written limits the value of this current study to the wider scientific community.

The authors would like to thank anonymous referee RC1 for the valuable insight. We very much agree with the referee's summary of the work's achievements and noteworthy findings. The authors are very supportive of the referee's opinion that there are substantial knowledge gaps in our current understanding of in-canopy processes. The authors also think that the present deposition modules coupled in the CTMs are lacking in simulating the processes and characteristics of the canopy surface.

In previous work, the authors noticed that the first version of the Noah-MP-WDDM model could not consider and express more precisely the stomatal resistance and leaf nitrogen physiology. Therefore, in this model description paper, the authors tried to combine several classical mechanisms with the previous version of the Noah-MP-WDDM model and carry out an elementary comparison of their performance.

As the referee mentioned, these mechanisms can not effectively solve the problem of inaccurate deposition velocity simulation. The authors are aware of this and agree with the referee. The current work is just the beginning in comparing different mechanisms in the Noah-MP-WDDM model, and the authors expact to identify areas where more research is needed based on the current attempt. We have reworked the scope and conclusions of the manuscript following the referee's suggestion. The authors have also included a the work plan for the next phase of the performance comparison with other deposition models and on other observation samples or land surfaces, as explained in the conclusion.

In addition, because we are not native speakers of English, the authors overstated language such as "emphasize importance," "best," and "better" in the first manuscript. Based on the suggestions of the referee, we have modified all such expressions in the hope of describing the results more accurately.

**The specific responses to the reviewers' comments are as follows:**
Specific comments:
Additional proof reading of the manuscript is needed.

L49. what is meant by this?
Thank you for pointing this out. What the authors mean here is that the calculation of these two resistances in different deposition mechanisms follows similar principles. In order to avoid ambiguity, the authors changed this sentence to the following: The calculation of these two resistances in different deposition mechanisms follows similar principles (Finnigan, 2000).

L64. I would like to see some additional citations here of modelling papers that represent Rc. (e.g. Wolfe et al., 2011, Delaria and Cohen 2020, Simpson et al., 2012, Ganzeveld et al., 2002, etc.).
Thank you for your advice. We carefully studied these references, cited them, and expanded the original description.

L 70. Would be good to discuss findings of the paper cited here, as the finding of Delaria and Cohen 2020 seem to tie in well with the purpose of this paper. This sentence as written just summarizes the introduction section of Delaria and Cohen 2020.
Thank you for your advice. Yes, the findings of Delaria and Cohen 2020 have important enlightening value for all existing land surface models. We tried to quote and tentatively comment on their findings; please see lines 75 to 79.

L 72. It seems you have missed a few recent papers that also support this finding and further discuss the compensation points and the roll of nitrogen availability on $NO_2$ uptake (Delaria et al., 2018 and 2020). Place et al, 2020 may also be interesting for you to look at.

Thank you for your suggestion. It is true that we had not read these two articles you mentioned when we started to improve the Noah-MP-WDDM model. Based on your suggestion, we have carefully read them. Their work is very in-depth and instructive for the discussion of nitrogen compensation points, and it is worthy of our consideration in further Noah-MP-WDDM model coupling work. Unfortunately, when we tried to read the code corresponding to the articles, we found that it was written using the Matlab language, and our model is based on the Fortran language, which makes it difficult for us to join and couple in our current work. It is expected that the code will be reconfigured in future versions so that it can be incorporated.

L101: why is it a better mechanism? Need citation for this? In what way is it "better"? Is this backed up by observational data?

Thank you for pointing this out. The authors agree that this expression is inappropriate, and it has been modified in the revised manuscript. The previous expression was based on the fact that the Ball-Berry scheme is based on experiments with net photosynthetic rate and stomatal conductance, while the Jarvis scheme is an empirical model of continuous multiplication that does not consider the physiological significance of parameters (Wang and Wen., 2010). In addition, although the Ball-Berry type stomatal resistance scheme behaves very similarly to the Jarvis type in modeling transpiration, the former scheme allows a direct coupling of terrestrial water and carbon fluxes and improves the simulation of vegetationatmosphere interactions (Niyogi et al., 2009, Yang et al., 2011). We tried to modify the entire subsection with more accurate expressions, avoiding irregular expressions such as "better" and "key." Please refer to lines 108 to 115 of the revised manuscript.

L106: One might argue that these "key plant physiological parameters" result in model overparameterizations are too species- and possibly individual- specific to be useful at a regional scale. How may spatial and species deviations in these parameters introduce uncertainties into your conclusions? What advantage or disadvantage does this have compared to the Jarvis model?

Thank you for your advice. Indeed, as you said, over-parameterization is prone to occur in the process of model characterization, especially on the regional scale. However, the problem with the original Noah-MP-WDDM model we used and improved is its insufficient ability to characterize the physiological processes of vegetation, rather than over-parameterization. At

present, the parameters we use are still the default parameters in Noah-MP's lookup table, and no experiment has been performed on its uncertainty. This suggestion is worthy of further work. There is indeed a risk to model performance due to parameter uncertainty. In line 112, the authors added a statement that these parameters may cause uncertainty, according to your suggestion. In addition, the authors believe that it is necessary to conduct a systematic assessment of the uncertainties caused by these parameters when we couple the current single-point Noah-MP-WDDM model into the regional model in the next step.

L115. Rns is not defined. Please define.

Thank you for pointing this out. The Rns is non-stomatal resistance. The definition was add to line 122 of the manuscript.

L 146: How is it simplified? Why is this simplification not ideal?

Thank you for pointing this out. As described in the subsequent line 137, although the default version of the NoahMP model sets a nitrogen-limiting factor, its calculation comes from two constants that make it impossible to characterize the changes in the nitrogen concentration on the leaf surface. Similarly, other land surface process schemes coupled in climate models and chemical transport models are basically the same, such as Noah, SiB, BATs, P-X, etc.

L150: I am confused. Are you stating that the nitrogen leaf content changes just the photosynthesis rate, or changes the relationship between Gs and nitrogen Vd? I would recommend looking at Delaria et al. and Place et al. 2020.

Thank you for pointing this out. According to the results of Zheng S. and Shangguan Z. (2007), for different types of vegetation, the effects of leaf nitrogen content have obvious differences in vegetation saturated photosynthetic rate ($P_{max}$), photosynthetic nitrogen use efficiency (PNUE), and quantum PS II electron transport ($\phi_{PSII}$). We quoted this in the original text to illustrate the possible impact of over-simplification by setting the leaf nitrogen content of different vegetation to a constant parameter in the Noah-MP land surface model. At the same time, the authors also recognize the results discussed in your suggested article, which point out that the relative importance of chemical sinks and deposition will vary with NOx concentration. The authors greatly appreciate the reviewer's recommendation of Delaria et al. (2020),

which is worthy of repeated study and quotation. We added a reference to it in the revised manuscript; please see line 160.

L198: Observations of deposition velocity are difficult and subject to different uncertainties based on observation method. How is Vd calculated in your observation data? What uncertainties may be present in the observation data? Citation to data?

Yes, the authors agree very much with your point of view; it is very difficult to carry out deposition observation. And there are different uncertainties based on observation method. However, in this paper, the authors main purpose is to use observations as the basis to explore differences in the simulation of deposition velocity by coupling different stomatal and photosynthetic mechanisms in the Noah-MP-WDDM model. The observed deposition velocity data used in this work came from Zhang et al., 2017, who also discuss on the sources of uncertainty in the observation itself. Vd was measured using the aerodynamic gradient method, which uses the concentration difference between the canopy interior and canopy exterior (Wu et al., 2015). The $NO_2$ concentration was measured using the $NO_2$ analyzer (Model T200, Teledyne-API, USA). The inspection instrument Model T200 used for observation had to run in a ventilated environment, so that the gas that had been assayed in the chamber could be discharged in time to ensure the accuracy of the next sample test results. But the instrument was limited by the conditions surrounding the flux tower; the assayed gas had accumulated (especially at night) in the reaction chamber, resulting in a partial (nocturnal) high observed value. In order to improve the reliability of the data, Zhang et al. excluded the abnormally high nighttime observation. Besides, the wind speed inside and outside the canopy is different and shows significant diurnal variation. During stable nighttime conditions, the uncertainty of the observation comes from the lack of turbulence. In order to ensure the reliability of the data, the observation data with small u* (smaller than 0.15 m s$^{-1}$) were also excluded in Zhang's study.

L208: Discussion of species present in the site considered is needed much earlier in the manuscript. Are you using species-specific parameters? If so, where are these parameters from and are there experimental data to support?

Thank you for your advice. For the model, the characterization parameters of forests are indeed very important. In this paper, the authors use the default lookup table of vegetation parameters from the Noah-MP land surface model, and do not specifically modify the parameters with the localized values of the observation area, which may also be a source of poor simulation performance. Thank you very much for this inspiration. We included this point of view in the corresponding position in the revised version of the paper. Please review it.

L214: Based on figure 2, even this scheme seems to have high stability of parameterization. It looks that it may even be anti-correlated with observed deposition velocity. From Figure 2 I would conclude that all mechanisms are bad at representing deposition velocity. It seems the goal of representing changes in Vd with different environmental conditions has failed, although diurnal cycles are captured seemingly better. I am curious what this data look like if you separate into different times of day (eg daylight hours only).

Thank you for pointing this out. First, as you said, based on the results in Figure 2, if you look at the simulation and observation results at all times together, all the mechanisms of the current version of the Noah-MP-WDDM model perform very poorly. However, from the characterization of the simulated diurnal variation in Figure 4, part of the simulation mechanism can show the relative day-night variation of the dry deposition velocity. It is worth emphasizing that in this manuscript, the authors are concerned about improving the function of the previous version of Noah-MP-WDDM and do not expect to solve all problems in one fell swoop. Second, the authors very much agree with the reviewer's suggestion of separating day and night. We tried to extract the results just during the day and the results just at noon. The results are shown in the figures below. As a supplement to Figure 2, it can be seen that for the deposition velocity during the daytime, the results do not show an obvious difference with the all-time results. But the just-noon results are seemingly captured better, and the differential performance of different test results can be displayed more clearly.

[Figure]

Figure 1: Comparison of observed and simulated $V_d$ of $NO_2$ during the day-time

[Figure]

Figure 2: Comparison of observed and simulated $V_d$ of $NO_2$ at noon

L225: I would like to see a time series of data over multiple days. Does your model not capture this turbulent exchange effect?

The time series of observation data and the three sets of simulation results of BN-11, BN-23, and BN-46 are shown in the figure below. It can be seen that part of the model results can characterize the effect of turbulent exchange. At the same time, it can be seen that, as mentioned in Zhang's article, there are many missing observation data. This lack of observation data also restricts our further evaluation of the improved performance of the model, so we can discuss and explain it only tentatively in this part of the article. We look forward to working with other observation teams to test and evaluate the simulation performance of our improved model.

[Figure]

Figure 3: Time series of observed and simulated $V_d$ of $NO_2$

L281. I don't think you can conclude from the results presented that the new v 1.42 is much improved. I don't see much evidence in the diurnal cycles that the agreement with observations is significantly (statistically) different, and there doesn't seem to be much evidence that would allow you to assert which model scheme is "best". The important conclusions from this work are in lines 287-291.

Thank you for your advice. The authors very much agree with the reviewer's view that lines 287-291 are important conclusions. For the work in the current manuscript, our main contribution is to improve the previous version of the Noah-MP-WDDM model and to tentatively compare the performance of various combinations of stomatal conductance schemes and nitrogen-limiting photosynthetic schemes and explain the simulation differences of the different schemes. Subject to the lack of observation data and uncertainty, we really cannot directly say whether a certain mechanism is the best. Therefore, the authors revised the conclusion of this paper after profound consideration and discussion based on the suggestions of the reviewer.

Figure comments:

Figures 4—7: How are errorbars calculated?

The errorbars are the standard deviation of the data. The authors use the $"varfun(@nanstd, time\_series\_data,' GroupingVariables',' Hours')"$ function of Matlab to create a vector of diurnal results and compute their standard deviation, excluding NaN values.

**RC2: 'Comment on gmd-2021-157', Anonymous Referee #1, 07 Jul 2021**

General comments

As a whole, the paper is interesting as it compares the performance of various combinations of stomatal conductance models and nitrogen-limited photosynthetic schemes. I think the paper should be published as it brings new and useful information to the scientific literature. However, the presentation has to be improved before publication. The authors should pay attention to details (see below). On the other hand, I am concerned about the fact that the authors did not show results of the Aphalo and Jarvis (1993) scheme (simulation BN-31-BN-33) on Figures 4-7 despite the fact that this scheme present among the lowest bias (MBM-3 in Figure 3). It would be interesting to include results from Aphalo & Jarvis model in Figs. 4-7 as well. Finally, the discussion section is absent and more interpretation of the results should be done including uncertainties and knwoledge gaps.

From the point of view of the English language, the authors should ask the help of a native English speaker to review and correct the manuscript because at times we see poor English. Moreover, proof editing is needed since there are a lot of typographical errors in the submitted manuscript.

The authors would like to thank anonymous referee RC2 for the valuable insight. We very much agree with the referee's summary of the work's achievements and noteworthy findings. The authors are very supportive of the referee's opinion that the paper should be published as it brings new and useful information to the scientific literature.

At the same time, the authors noticed that for the two combinations BN-13 and BN-33 with lower bias (-0.0187) mentioned by the reviewer, the results are very close to the lowest bias result (BN-23, -0.0185), and their daily change is basically similar, with only a small difference in Rb and the standard deviation (as shown in the figure below). So it was not highlighted in the discussion of the previous version. The authors added an explanation of the results of BN-13 and BN-33 in line 215 of the main text; please see the revised version.

In addition, because we are not native speakers of English, the authors asked a native English speaker to review and correct the manuscript. Based on the suggestions of the referee, we have modified all such expressions, hoping to describe the results more accurately.

Specific comments

- In the methodology section it is necessary to make clear that model output interpolated at one point (at the measurement tower site) is not necessirily comparable with a measurement point. - A discussion section should be added to discuss several weakness and uncertainties of inputs and results. For exemple, tower measurement is made at one location not necessary spatially representative of a whole model grid tile (whereas model output are average over a bigger area). The authors should also mention the resolution (grid spacing) of the model used and should discuss the validity and uncertainties of measurement versus model. What about scale dependency of dry deposition ?. A discussion section should be added to review the results and provide further interpretation and describe uncertainties and knowledge gaps in a better way.

Thank you for your advice. The authors fully understand what the reviewer mentioned. For the land surface model, in general, the grid point center subject to it does not necessarily match the location of the observation site, and therefore the comparison with the observation site results does not need to be very accurate. The authors very much agrees with the reviewer that this scale conversion will bring a certain uncertainty, especially when a single-point test model is coupled with a regional or global model. This is very important and worthy of discussion. However, in this paper, we use the single-point Noah-MP-WDDM model to improve and couple the mechanisms that affect dry deposition simulation. The driving data come from regular meteorological observation data at the observation station, and the grid and station location mismatch or the spatial interpolation step does not exist in the current paper, so the authors did not discuss it in the original text. Based on the reviewer's suggestions, the authors revised the methods and conclusions of the paper, and further clarified the model setting. The authors expect to couple the current improved mechanisms to a regional and global land surface model in the next step, and it is hoped that an in-depth discussion will be conducted on the impact of this scale-up effect.

-Lines 93-97: not clear to what refers option 1. For those not very familiar with this model, it is hard to follow. More details are needed concerning options description and model characteristics.

Thank you for pointing this out. Yes, this is the authors' negligence. All the default parameterization options are used for these schemes, and the description has been revised in the paper. On this basis, the authors also revised the description of the other schemes in lines 96-97. Please see the revised version.

-Section 2.2 Coupling of stomatal resistance scheme. Not clear how the coupling is done. Please explain.

Thank you for pointing this out. The integration of these schemes is done by writing the equations from Table 1 as subroutines in the Noah-MP-WDDM model. The authors first commented about the original default equation, found the variables needed in the coupled equations from the original source code, used the variables as input to the corresponding equation, and then used the output variable result in the subsequent calculation. Some variables that are not in the original source code are calculated indirectly through other related variables. At the same time, in the process of this equation coupling, it is also necessary to declare the data type, calling listcalling tree, and to set the input and output defining declaration of the corresponding variable.

-Stomatal resistance or stoma resistance ? Throughout the document you should use stomatal resistance or stomatal conductance. E.g. line 13 please replace stoma resistance  stomatal resistance

Thank you for pointing this out. All instances of "stoma" have been replaced with "stomatal" in the manuscript.

-In models of stomatal conductance, Gs = k A*RH/[CO2], inputs for RH and A are available but how did you obtain [CO2] ?. By measurement ? CO2 is usually not available from numerical models. Please clarify.

Thank you for this question. The concentration of $CO_2$ is usually a variable in climate or atmospheric models, but as you mentioned, it is generally not available in quite a lot of land surface or biosphere models. In the current version of the single-point Noah-MP-WDDM model, the concentration of $CO_2$ is just a parameter input. After discussion, the authors believe that this is also one of the reasons for the small difference in the simulation results of the different schemes. We hope to be able to test the $CO_2$ monitoring data as different parameter input for the model from observation networks such as Fluxnet or ChinaFlux in the future. Thank you for your reminder.

- Line 233-234 smallest simulation deviation. This is poor English.

Thank you for pointing this out. The original intention of the authors here was to express it as the "minimum bias." Based on the suggestions of the referee, we have modified all such expressions, hoping to describe the

results more accurately.

-The reference to simulation BN1-BN76 should also be better explained there in reference to Table 3. I suggest putting the name of the simulation (BNx) in Figure 3 (just below the bias values). It would become clearer for the reader and allow a direct comparison and consistency with the following figures 4-7 where simulation name BN are used.

Thank you for your suggestion. However, it is difficult to add references to each mechanism in Table 3. We tried to modify the form of Figure 3 with your next suggestion, as shown in the figure below. And as you mentioned, it becomes clearer for the reader and allows a direct comparison and consistency with the following figures 4-7 where the simulation name BN is used.

[Figure]

Figure 4: Mean bias of observed and simulated $V_d$

-Table 3 and Figure 3 are presented in different order (columns and rows are inverted). Please transpose the matrix in Table 3 to be consistent with Figure 3.

Thank you for your suggestion. As shown in the figure above, the content of Table 3 has been fully expressed in the new Figure 3, so we deleted the original Table 3 to avoid confusion.

-Figures 4-7. Why not putting results of simulation from Aphalo and Jarvis (1993) model in the figures (Sim BN31-BN36) ? It would have been interesting to present also simulation of BN31-BN33 in your figure since they show the lowest bias in Figure 2.

Thank you for pointing this out. First of all, the different presented order of the authors' presentation of the tables and figures made the reviewers mistakenly believe that the low bias simulated combination in Figures 2 and 3 was BN-31 $\sim$ BN-33. In fact, it should be BN-13 $\sim$ BN-33, where BN-23 has the lowest bias. Second, as shown in the figure below, we display the simulated diurnal variation results of these cases in the form of Figures 4$\sim$7. It can be seen that the results of using the BN-13 and BN-33 mechanisms are basically the same as the results of BN-23. The discrimination causes only a slight disturbance to Rb. This may be due to the uses of $CO_2$ concentration as a parameter input to constrain the performance of the model. On the other hand, the differences of MBM1 to MBM3 are masked by the MNM process.

[Figure]

Figure 5: Diurnal variation of observed and simulated $V_d$ with BN-13 and BN-33 highlighted

[Figure]

Figure 6: Diurnal variation of observed and simulated $R_a$ with BN-13 and BN-33 highlighted

[Figure]

Figure 7: Diurnal variation of observed and simulated $R_b$ with BN-13 and BN-33 highlighted

[Figure]

Figure 8: Diurnal variation of observed and simulated $R_c$ with BN-13 and BN-33 highlighted

Minor

Line 9 and 11 Nitrogen-limitings -¿ nitrogen-limiting
Thank you for pointing this out. It has been modified.

Line 34 deviation - ??? standard deviation ?
Thank you for pointing this out. It has been modified. The deviation here means "uncertainty." When we tried to use Google Translate to English, its priority translation became "deviation."

Lines 42 and 44 typo. Please correct.

Thank you for pointing this out. It has been corrected.

Lines 45 Equ. - Eq.
Thank you for pointing this out. It has been corrected.

Figure 5,6 and 7. Units needed in the legend (s/cm ? or s/m?). Please specify.
Thank you for pointing this out. The units are s/m and they are added to the Y-labels.

Line 169, 170. 212 Typos. Please correct.
Thank you for pointing this out. It has been corrected.

Line 205 typo. 0.05 cms  0.05 cm/s
Thank you for pointing this out. It has been corrected.

Line 215, 234, etc.: deviation ? you mean standard deviation of what ? please clarify.
Thank you for pointing this out. The "deviation" in lines 215 and 234 should be "bias." Again, the Google Translate priority translation was "deviation." We have asked a native English speaker to help us polish the manuscript during the revision process.

Line 216 need space between number and units (typo)
Thank you for pointing this out. It has been corrected.

Lin 226 English could be improved, e.g. The black line in Fig 4 is BN-11 and the green line is BN-23  The black line in Fig. 4 corresponds to experiment BN-11 and the green line to BN-23, respectively.
Thank you for giving an example. It has been improved.

Line 227 This sentence does not make sense. Vd simulation upward  simulated Vd values are increased.

Thank you for pointing this out. It has been improved.

Line 227-230 badly worded. Please re-write.
Thank you for pointing this out. It has been rewritten.

Line 233 smallest deviation (standard deviation ?)
Thank you for pointing this out. It should be "minimum bias."

Line 234 bias 0.001 cm/s ? where does it come from ?
This bias comes from all BN-46 midday simulated results and all the observed values at noon.

Line 245 s/m or s/cm ? in other parts of the manuscript s/cm units are used. Why here use s/m ? Please be consistent with units used elsewhere in thee document. Do not mix units it is confusing for the reader. Either s/m in the whole document or s/cm but not both.
Thank you for pointing this out. The units of resistances output by Noah-MP-WDDM are "s/m," while the unit of deposition velocity is "cm/s." We have checked and unified all the units in the whole manuscript.

Section 3.4.1 badly written
Thank you for pointing this out. It has been rewritten.

Line 126. Later, the method of Lohammar et al (1980). This does not make sense. The authors just previously talk about a reference Leuning (1990). Year 1980 comes before 1990 not later. This poor English. Please reword.
Thank you for pointing this out. The situation here is that the method of Lohammar et al (1980) was applied by Leuning (1995)'s work. It is subject to poor English proficiency, which we have not stated clearly before. This part has been rewritten.

---

## Author Response (AR2)

**Reply to 'Comment on gmd-2021-157'**

December 3, 2021

Dear Editor, Thank you for your review and comments on our manuscript. We appreciate your insightful constructive suggestion and expand the discussion following your suggestion. The changes have been highlighted in blue and red fonts in the track-change file by using the LaTeXdiff tool. We sincerely hope that the manuscript in its revised form will satisfy the queries of the referee and will be accepted for publication.

The comments from editor and referee are colored black.
The authors' replies are colored blue.

**Comments to the author & review by editor**

Dear Authors,

I apologize for the long delay in getting back to you. The second review from one of the reviewers never came back. One of the main comments from this reviewer was the lack of discussion in your manuscript. The reviewer requested an expanded discussion and conclusion, and I agree. I have reviewed the revised manuscript, and i do not think that your edits would have satisfied the reviewer. You have not really expanded much on the discussion and put your work in context of existing literature. Your edits are fairly small, i don't see much expansion at all.

Hence my request is to re-consider this properly. I think you can do a much better job at expanding the discussion and conclusion. It might be helpful to read the discussion sections of good papers in this field, to get a better idea of how to better craft your discussion and conclusion. It is currently rather short. Please give this due consideration in your re-submission.

Kind regards and apologies again for the long delay,
Jatin

Dear Editor, Thank you for your review and comments on our manuscript. We appreciate your insightful constructive suggestion and expand the discussion following your suggestion. The changes have been highlighted in blue and red fonts in the track-change file by using the LaTeXdiff tool. We sincerely hope that the manuscript in its revised form will satisfy the queries of the referee and will be accepted for publication.

The main change of this revised manuscript is add a new section to discuss the current knowledge gap on the relationship of the in-canopy process and the dry deposition simulation. We summarized some point from the previous comments suggested by referee and the replies to referee. We also read further literature and re-consider about the importance of the in-canopy dry deposition process. We insist the consideration of biological physiological processes inside the vegetation canopy needs to be continuously refined while the understanding of the land-atmosphere exchange process in the ecosystem grows.

In addition, the comparison with other models and observations are discussed in the fourth paragraph of this section. It is found that the dry deposition velocities of $NO_2$ range obtained by most of the model results is basically lower than the observed value obtained by the eddy correlation method, which is relatively consistent with the performance of most of the mechanism simulation results in this study. The limitations and the uncertainties issues of parameters are also discussed in the last paragraph of this section. We believe that it is necessary to further carry out the measurement and accurate characterization of model parameters.

**RC2: 'Comment on gmd-2021-157', Anonymous Referee #2**

Recommendation to the editor
1) Scientific significance
Does the manuscript represent a substantial contribution to modelling science within the scope of this journal (substantial new concepts, ideas, or methods)?
Good
2) Scientific quality

Are the scientific approach and applied methods valid? Are the results discussed in an appropriate and balanced way (consideration of related work, including appropriate references)? Do the models, technical advances and/or experiments described have the potential to perform calculations leading to significant scientific results?

Good

3) Scientific reproducibility

To what extent is the modelling science reproducible? Is the description sufficiently complete and precise to allow reproduction of the science by fellow scientists (traceability of results)?

Good

4) Presentation quality

Are the scientific results and conclusions presented in a clear, concise, and well structured way (number and quality of figures/tables, appropriate use of English language)?

Good

For final publication, the manuscript should be

accepted as is

Were a revised manuscript to be sent for another round of reviews:

I would not be willing to review the revised manuscript.

The authors would like to thank anonymous referee RC2 for the previous valuable comments and the suggestions of our work's achievements and noteworthy findings. The authors are very supportive of the referee's opinion that the paper should be accepted as it is.

---

## Author Response (AR3)

**Reply to 'Comment on gmd-2021-157'**

December 9, 2021

Topical Editor decision: Publish subject to technical corrections by Jatin Kala

The comments from editor and referee are colored black.
The authors' replies are colored blue.

**Comments to the author**

Dear Authors,

Thanks for expanding on the discussion. Please consider the following minor edits:

Page 11, line 318, "and that the results" probably should be "and our results"? Please revise.

Line 325, I suggest "non-stomatal and stomatal pathways".

Line 328, experiments cannot "believe" anything. Rather i assume you mean "assume" rather than "believe"?

Line 345, "These" rather than "They"?

Lines 357 to 358 - use of the "on the other hand" is repetitive, please revise.

Dear Editor,

Thank you for review and accept our manuscript. We appreciate your insightful constructive suggestion for pointing these expression questions. Because we are not native speakers of English, the accuracy of our statements needs to be further strengthened. we have modified all such expressions following your suggestions. The changes have been highlighted in blue and red fonts in the track-change file by using the LaTeXdiff tool.

Thank you very much for your cooperation.